# Role of Some Food-Grade Synthesized Flavonoids on the Control of Ochratoxin A in *Aspergillus carbonarius*

**DOI:** 10.3390/molecules24142553

**Published:** 2019-07-13

**Authors:** Alessandra Ricelli, Martina De Angelis, Ludovica Primitivo, Giuliana Righi, Carla Sappino, Roberto Antonioletti

**Affiliations:** 1Institute of Molecular Biology and Pathology-CNR P.le Aldo Moro 5, 00185 Rome, Italy; 2Department of Chemistry, Sapienza University, P.le A Moro 5, I-00185 Rome, Italy

**Keywords:** flavones, aromatic hydroxylation, *Aspergillus carbonarius*, ochratoxin A, lipoxygenase

## Abstract

Ochratoxin A (OTA) is a mycotoxin with a serious impact on human health. In Mediterranean countries, the black Aspergilli group, in particular *Aspergillus carbonarius*, causes the highest OTA contamination. Here we describe the synthesis of three polyphenolic flavonoids: 5-hydroxy-6,7-dimethoxy-flavone (MOS), 5,6-dihydroxy-7-methoxy-flavone (NEG), and 5,6 dihydroxy-flavone (DHF), as well as their effect on the prevention of OTA biosynthesis and lipoxygenase (LOX) activity in *A. carbonarius* cultured in a conducive liquid medium. The best control effect on OTA biosynthesis was achieved using NEG and DHF. In fungal cultures treated with these compounds at 5, 25, and 50 μg/mL, OTA biosynthesis significantly decreased throughout the 8-day experiment. NEG and DHF appear to have an inhibiting effect also on the activity of LOX, whereas MOS, which did not significantly inhibit OTA production, had no effect on LOX activity. The presence of free hydroxyls in catecholic position in the molecule appears to be a determining factor for significantly inhibiting OTA biosynthesis. However, the presence of a methoxy group in C-7 in NEG could slightly lower the molecule’s reactivity increasing OTA inhibition by this molecule at 5 μg/mL. Polyphenolic flavonoids present in edible plants may be easily synthesized and used to control OTA biosynthesis.

## 1. Introduction

Ochratoxin A (OTA) is a secondary metabolite, toxic for humans and animals, which is synthesized by several fungi of the Aspergillus and Penicillium species. Such fungi are frequently encountered as contaminants of many foodstuffs; in particular, Penicillium species are found more commonly in cold climates and Aspergillus species more frequently in warm climates [1].

OTA has various acute and chronic toxic effects [2] on humans and animals and has been classified as a group 2B by the International Agency for Research on Cancer (IARC) [3]. Consequently, thresholds for its maximum presence in foodstuffs have been established by most governmental jurisdictions, e.g., the European Union [4]. Acute mycotoxicoses have a rapid onset and an obvious toxic response, whereas chronic toxicity is characterized by low-dose exposure over a long period and also results in serious toxic effects [5].

Worldwide, approximately 25% of food crops are affected by mycotoxins causing a loss of nearly 10^9^ tons of foodstuff per year [6,7]. Among the occurring mycotoxins, OTA is found in varying amounts in a wide variety of commodities such as raisins, barley, soy products, and coffee [6].

OTA contamination’s economic costs result mainly from yield loss, losses in livestock and animal productivity, reduced value of raw food materials when contaminated below the tolerated limit, and trade impacts [8]. Further costs include expenditures for implementation of sampling and analyses, to run the regulatory programs organized by the dedicated agencies, and segregation and disposal of contaminated food [9].

Although the decontamination of mycotoxins present in feeds or foods somehow may be possible, the elaboration of a strategy to prevent the biosynthesis of mycotoxins in the first instance is the priority. Unfortunately, attempts to control OTA contamination by means of continuous surveillance of agricultural crops, animal feedstuffs, and human foods through the entire production chain, have been largely unsuccessful. Oxidative stress has been shown to play a crucial role in both OTA biosynthesis and OTA toxicity; in particular, the significant role of lipoxygenase (LOX) activity has been observed in some OTA producing Aspergillus [10].

Several studies have been performed using antioxidant molecules [11,12,13], a still promising research path. Antioxidants can both counteract the adverse effects of reactive oxygen species (ROS) generated under OTA contamination and control the biosynthesis of the toxin itself [14,15]. Given that in most cases OTA biosynthesis takes place in food raw materials and foods, many studies of this contaminant’s prevention have evaluated extracts from plants that are edible or otherwise already known for their beneficial health properties [16].

In order for the bioactive molecules present in such plants to be studied, they must be extracted with solvent mixtures or synthesized, depending on the balance of each protocol’s costs and benefits. Flavonoids are among the most important bioactive molecules identified in the extracts of plants known to popular medicine. Several studies have highlighted their bioactive potential, and, in particular, their antioxidant, anti-inflammatory, and antiviral activities [17,18,19,20].

In this study, we considered three polyphenolic flavones: 5-hydroxy-6,7-dimethoxy-flavone (mosloflavone, MOS), 5,6-dihydroxy-7-methoxy-flavone (negletein, NEG), and 5,6-Dihydroxy-flavone (DHF) (Figure 1).

The first two compounds are present in some plants used in folk medicine, such as *Centaurea clementei* (Boiss) (negletein) [21], and *Desmos chinensis* (Lour.) (mosloflavone) [22], their antioxidant characteristics are well known and several studies have found that they have beneficial effects on human health [23]. MOS and NEG were obtained by a simple synthesis starting from the 7-methyl ether of crysin, similarly to what was described in Righi et al. [24]. The third compound (DHF), as far as we know, has not been detected in nature and is not a commercial product, but is structurally related to NEG and MOS and has been obtained through a specific synthetic protocol [25]. The only available studies of DHF’s biological activity are, to our knowledge, that conducted by Gao et al. [26] testing the effect of hydroxyl substituents at positions 5, 6, or 7 in the A-ring of the 5,6,7-trihydroxyflavone on rat intestinal alfa-glucosidase inhibition, and that conducted by Lombardo et al. [27] investigating the cytotoxic and antioxidant activities of baicalein, mosloflavone, negletein, and 5,6-dihydroxyflavone at very low and physiologically relevant levels, on two different cell lines, L-6 myoblasts and THP-1 monocytes.

In our study, MOS and NEG, were synthesized from 7-methyl ether of crysin by means of a quick and easy synthetic method, previously described in [24]. DHF was synthesized from 6-hydroxy-flavone by using the synthetic method reported in [25]. MOS, NEG, and DHF were subsequently used to treat OTA-producing *Aspergillus carbonarius* cultures. The effect of the synthesized molecules on OTA biosynthesis, fungal growth, and LOX activity were investigated.

## 2. Results

We prepared MOS **4** and NEG **5** by a simple synthesis consisting of two steps to introduce the oxygenated moieties into the activated aromatic A-ring; a bromination followed by a substitution and a methanolysis protocol on 5-hydroxy-7-methoxy-flavone (7-methyl-ether of crysin) **1**. DHF **7** was obtained by a direct hydroxylation of the aromatic A-ring of 6-hydroxy-flavone **6**.

### 2.1. Synthesis of Mosloflavone 4, Negletein 5, and 5,6-Dihydroxy-flavone 6

The bromination step on **1** was performed by tetrabutylammonium tribromide (TBATB) in chloroform at room temperature; this reagent showed a very high efficiency but a low regioselectivity (Scheme 1). In fact, we obtained in good yields, a mixture (about 1:2 respectively) of two regioisomers: The monobromo derivatives **2** (5-hydroxy-7-methoxy-8-bromo-flavone) and **3** (5-hydroxy-6-bromo-7-methoxy-flavone). The two bromides were not separable by common chromatographic techniques, their structures were demonstrated previously by Righi et al. [24]. In this study we used the mixture of bromides **2** and **3** to obtain directly by methanolysis the molecule **4**.

The methanolysis consisted of mixing at room temperature a solution of NaOMe in MeOH with a suspension of CuBr in dimethylformamide (DMF), the obtained mixture was added to the solution of the aromatic bromides (**2** + **3**) in DMF at 120 °C. This reaction is regiospecific and leads only to one product: The 5-hydroxy-6,7-dimethoxy-flavone **4** (mosloflavone). The substitution reaction can occur, in good yield, as reported in [24].

Subsequently the mosloflavone **4** was demethylated in C-6 by reaction with hydrobromic acid and acetic acid, at reflux conditions, and, in this manner, we prepared the 5,6-dihydroxy-7-methoxy-flavone **5** (negletein) in high yield (Scheme 1).

The third molecule to be tested was **7** (5,6-dihydroxy-flavone), which was synthetized from 6-hydroxy-flavone **6** by the ipervalent iodine oxidant 2-iodoxybenzoic acid (IBX) as reported in [25]. In this oxidative reaction, IBX proved to be an excellent reagent for a highly regioselective aromatic hydroxylation of monohydroxylated-flavone, generating the corresponding cathecolic system as illustrated in Scheme 2.

### 2.2. Effect of MOS, NEG, and DHF on A. carbonarius Growth and OTA Biosynthesis

To evaluate the influence of MOS, NEG, and DHF on *A. carbonarius* growth and OTA production, each compound was added at concentrations ranging from 5 to 50 µg/mL in Erlenmeyer flasks (50 mL) filled with 25 mL of synthetic liquid medium Czapek-Dox plus yeast extract 0.5% (CDY 0.5%) and inoculated with 3 × 10^5^ conidia. The selected medium is conducive for both *A. carbonarius* growth and OTA biosynthesis. The evaluation of *A. carbonarius* growth and OTA production trend was conducted after 3, 5, and 8 days of incubation at 25 °C corresponding to the beginning and the end of the exponential growth phase, and to the onset of the plateau phase, respectively (Figure 2).

The amount of OTA found in the cultural filtrate of the untreated cultures was 7, 98, and 209 ng/mL after 3, 5, and 8 days respectively. In *A. carbonarius* cultures treated with MOS at any of the tested concentrations, the biosynthesis of OTA was not significantly affected throughout the duration of the experiment. When the fungus was treated with NEG, OTA production was significantly controlled. In particular, a reduction in OTA production of more than 60% compared to untreated cultures (Cont) was observed even when NEG was added at only 5 µg/mL, and a reduction of more than 90% was obtained when NEG was added at 50 µg/mL. These reduction levels were maintained throughout the duration of the experiment. The addition of DHF at 5 µg/mL to the culture of *A. carbonarius* led to a reduction of toxin concentration of about 30% only in the early phases of the experiment (after a 3-day incubation). When incubation times were more protracted (5 or 8 days) the compound completely lost its effect. Instead, when DHF was used at 25 µg/mL or 50 µg/mL, OTA production was about 90% inhibited after 3 days of incubation; even at 8 days of incubation, the inhibitive effect was maintained at the same level. None of the tested compounds had any effect on fungal growth (data not shown).

### 2.3. Effect of MOS, NEG, and DHF on LOX Activity in A. carbonarius

We verified LOX activity in *A. carbonarius* cultures treated with MOS, NEG, and DHF (Figure 3a–c). The analysis was accomplished by measuring the absorbance units/mg protein (Abs U/mg protein). LOX is capable of producing allyl hydroperoxides from polyunsaturated fatty acids that have a system of alternating *cis* double bonds. Such hydroperoxides can induce the biosynthesis of several mycotoxins in different fungi, e.g., aflatoxins, sterigmatocistin, and OTA produced by *A. flavus*, *A. nidulans* and *A. carbonarius* [28]. NEG and DHF, which are effective at inhibiting the biosynthesis of OTA in the assayed *A. carbonarius* isolate, also inhibit LOX activity, whereas MOS, which did not significantly inhibit OTA production, did not.

## 3. Discussion

Different flavonoids are accumulated in plant tissues as a response to biotic and abiotic stress such as pathogen and insect attack, UV radiation, and wounding [29].

They have been reported to play a variety of roles in plants, such as UV protection, pigmentation, growth regulation, stimulation of nitrogen-fixing nodules, and disease resistance [30,31].

Some flavonoids, such as MOS and NEG, have been extensively studied mainly due to their promising antimicrobial, antitumor, and neuroprotective activity [20,23,32]. However, little information is available on other molecules, despite their structural relation to MOS and NEG; for example, few studies have been done on 5,6-dihydroxy-flavone **7** [26,27]. In any case, thus far none of the three molecules considered here has been evaluated as possible means of countering OTA biosynthesis.

We prepared MOS **4** and NEG **5** from crysin-7-methyl-ether **1** by the synthetic process described in [24] that consisted of a simple two-step synthesis (bromination and methanolysis), in order to introduce the oxygenated moieties into the activated aromatic A-ring (Scheme 1). However, we modified the protocol by skipping the acetylation of the bromide mixture (**2** and **3**) since the two regioisomers were already characterized [24]. Therefore, we directly submitted the mixture to methanolysis, obtaining as the only product the 5-hydroxy-6,7-dimethoxy-flavone **4** (MOS). Through careful, fine management of the demethylation reaction time we selectively obtained the product of mono-demethylation **5**. This enabled a high yield (82%) while shortening the process. NEG was easily obtained through MOS demethylation (94% yield from MOS). DHF was obtained in high yield (96%) by a direct hydroxylation of the aromatic A-ring following the protocol described in [25] which is the best assessed synthetic process in terms of yield (Scheme 2).

The timeframe of the evaluation of *A. carbonarius* growth and OTA production trends (with 3, 5, and 8 days of incubation) has allowed us to monitor both the progress of fungal growth (primary metabolism) and the possible production of OTA (produced by secondary metabolism). MOS’s scarce and transient capacity to inhibit OTA biosynthesis could be due to the presence of two methoxy groups (C-6 and C-7) in the molecule. In particular, the second methoxy group’s presence in C-6 dramatically lowers the molecule’s reactivity. MOS’s poor reactivity does not allow the molecule to neutralize ROS that are formed, for example, as a result of LOX activity (Figure 3a). The correlation between the inhibition of LOX activity and the decrease of OTA production has been reported in *A. ochraceus* [10] and in *A. carbonarius* [28].

The cultures treated with NEG significantly inhibited OTA production (Figure 2). In particular, a significant and long-lasting effect is evident starting from treatment at 5 µg/mL, a result due perhaps to two free hydroxyl groups in a catecholic position (C-5, C-6).

A remarkable inhibition is evident when DHF is added to the fungal culture. In particular, it lasts for only 3 days when DHF is added at 5 µg/mL, but is both more inhibitive and longer lasting when DHF is added at 25 µg/mL and 50 µg/mL (Figure 2). The inhibition of OTA biosynthesis can be correlated with the inhibition of LOX activity (Figure 3a–c). This could result from two free hydroxyl groups’ presence in cathecolic position in DHF, which may render the molecule so highly reactive that its antioxidant effect is quickly exhausted if the concentration is not high enough.

Both NEG and DHF have a catecholic position (C-5, C-6), but NEG’s C-7 position is occupied by a methoxy group. This substitution could slightly lower the reactivity of the molecule preventing NEG from reacting too hastily with the radicals. This could account for the better OTA inhibition of NEG at 5 µg/mL compared to that of DHF at the same concentration, that probably reacts too quickly with ROS.

The presence of free hydroxyl in catecholic position in the molecule appears to be a determining factor for exerting a significant inhibitory effect on OTA biosynthesis, but the methoxy group’s presence in C-7 also appears to influence the considered biological system. MOS’s lack of efficacy therefore could be due to the presence of a further methyl ether in C-6 which impairs the cathecolic structure.

## 4. Materials and Methods

The solvents and the reagents used in this study were purchased by Sigma-Aldrich S.r.l. (Milan, Italy). The solvents were HPLC grade. Flash column chromatography was conducted on Silica gel 230-400 mesh (Merck S.p.A., Milan, Italy). Reactions were monitored by TLC using Merck silica gel 60F-254 plates and visualized under UV light at 254 λ and/or by phosphomolybdic acid (10% sol in EtOH).

NMR spectra were recorded on a Mercury 3000 instrument (Varian, Palo Alto, CA, USA) (^1^H, 300 MHz; ^13^C, 75 MHz). Chemical shifts were calculated from the residual solvent signals of δ_H_ 2.04 ppm and δ_C_ 206.0 ppm in acetone-*d*_6_, δ_H_ 7.24 ppm and δ_C_ 77.0 ppm in chloroform-*d*_1_, δ_H_ 2.49 ppm and δ_C_ 39.5 ppm in DMSO-*d*_6_. Melting points were measured on a FP80 instrument (Mettler-Toledo, Columbus, OH, USA) and were uncorrected. Synthetic media for fungal growth were purchased by Difco (Becton, Dickinson and C., Franklin Lakes, NJ, USA).

### 4.1. Synthesis of 5-hydroxy-7-methoxy-8-bromo-flavone 2, and 5-hydroxy-6-bromo-7-methoxy-flavone 3

TBATB (1590 mg, 3.73 mmol) was added in one portion to a solution of **1** (1000 mg, 3.73 mmol) in chloroform (14 mL). The mixture was left stirring at room temperature for 2 h, and then the mixture was diluted with water (30 mL) and extracted with ethyl acetate (3 × 30 mL). The extracts were washed with brine (2 × 20 mL), dried over anhydrous Na_2_SO_4_, and the solvent evaporated under reduced pressure. After purification by chromatography on silica gel a 1:2 mixture of **2** and **3** (1204 mg, 3.47 mmol, 93%) was obtained as yellow powder. The two isomers were inseparable by common techniques and were utilized in the next reaction without further purification.

### 4.2. Synthesis of 5-hydroxy-6,7-dimethoxy-flavone 4 (MOS)

To a suspension of CuBr (330 mg, 2.31 mmol) in DMF (7.0 mL) a 25% solution of NaOMe in methanol (21.0 mL, 92.33 mmol) was added at room temperature and left under stirring until a bright blue color appeared (about 1 h). This solution was added to a mixture of two bromo-phenol compounds **2** and **3** (1000 mg, 2.88 mmol) in DMF (9.5 mL) at 120 °C in 2 mL portions. The mixture was left stirring for 5 h, then cooled to room temperature, quenched with a cold 2 M solution of HCl in water (40 mL) and extracted with ethyl acetate (3 × 50 mL). The extracts were washed with brine (3 × 50 mL), dried over anhydrous Na_2_SO_4_, and the solvent evaporated under reduced pressure. Compound **4** (710 mg, 2.38 mmol, 82% yield) was obtained, as yellow powder, after chromatography on silica gel (hexane/ethyl acetate 8:2). Data agreed with those reported in literature [26]. M.p.: 155–159 °C. ^1^H-NMR (chloroform-*d*_1_) δ (ppm): 12.67 (1H, s, C5-OH), 7.87–7.84 (2H, m, C2′-H, C6′-H), 7.55–7.53 (3H, m, C3′-H, C4′-H, C5′-H), 6.66 (1H, s, C8-H), 6.55 (1H, s, C3-H), 3.95 (3H, s, CH_3_O), 3.90 (3H, s, CH_3_O). ^13^C-NMR, 182.4, 164.2, 159.3, 153.2, 153.0, 132.4, 131.7, 131.1, 128.7, 125.9, 106.0, 105.4, 90.1.

### 4.3. Synthesis of 5,6-dihydroxy-7-methoxy-flavone 5 (NEG)

A solution of **4** (600 mg, 2.00 mmol) in acetic acid (37 mL) and hydrobromic acid (15.2 mL, 47% in water) was refluxed for 3 h, then the solution was cooled to room temperature and poured onto ice. The resulting precipitate was filtered, washed with water (3 × 15 mL), and dried in the oven (60 °C) overnight. Compound **5** (542 mg, 1.91 mmol, 95% yield) was obtained as yellow powder. M.p. 235–238 °C. ^1^H-NMR (dimethylsulfoxide-*d*_6_) δ (ppm): 12.47 (s, OH-5), 8.05 (s, OH-6), 7.87 (m, 2H, C2′-H, C6′-H), 7.56 (m, 3H, C3′-H, C4′-H, C5′-H), 6.98 (s, 1H, C8-H), 6.79 (s, 1H, C3-H), 3.14 (3H, s, OCH_3_). ^13^C-NMR (dimethylsulfoxide-*d*_6_) δ (ppm): 182.5, 164.0, 153.3, 151.2, 146.1, 132.1, 130.1, 129.2, 126.5, 126.4, 105.8, 105.6, 90.6. Other data agreed with those reported in [22].

### 4.4. Synthesis of 5,6-diydroxy-flavone 7 (DHF)

To a stirred solution of 5-hydroxy-flavone **6** (510 mg, 2.0 mmol) in DMSO (20 mL) was added, in portionwise, 1.2 eq of IBX (680 mg, 2.4 mmol), prepared as described in [33]. The reaction was stirred at room temperature for 1 h, during the reaction a chromatic change from yellow to brown was observed. At the end, H_2_O (20 mL) and an excess of Na_2_S_2_O_3_ were added, and the solution was stirred until it returned yellow. Ethyl acetate (30 mL) was added to the mixture, which was then treated with pH 8.5 buffer solution to remove *o*-iodobenzoic acid. The aqueous phase was extracted with ethyl acetate (3 × 20 mL). The collected organic phases were washed with a brine (3 × 20 mL) and dried over Na_2_SO_4_. After evaporation in vacuum of the solvent, the product was isolated by chromatographic purification on silica gel. The 5,6-diydroxy-flavone **7** was obtained as yellow solid (487 mg, 1.92 mmol, 96% yield). Data agreed with those reported in literature [26]. M.p. 190–193 °C. ^1^H-NMR (methanol-*d*_4_) δ (ppm): 7.90 (m, 2H, C2′-H, C6′-H), 7.55–7.45 (m, 3H, C3′-H, C4′-H, C5′-H), 7.26 (d, 1H, *J* = 8.9 Hz, C7-H), 7.00 (d, 1H, *J* = 8.9 Hz, C8-H), 6.71 (s, 1H, C3-H). ^13^C-NMR (methanol-*d*_4_) δ (ppm) 184.1, 164.8, 150.1, 145.3, 140.0, 131.9 131.4, 129.0, 126.4, 121.7, 111.0, 106.8, 105.0.

### 4.5. Fungal Cultures

An *A. carbonarius* isolate from wine grapes grown in the countryside of Manduria (Taranto, Italy) able to produce high concentrations of toxin (2.0–2.5 µg/mL medium) under the experimental conditions was used. The isolate was maintained in Czapek-Dox Agar (CDA) slants at 25 °C, and, after 10 days, conidia were scraped off the cultures and put in sterile distilled water. The conidia suspension was counted by a Thoma counting chamber (Merck KGaA, Darmstadt, Germany) and 3 × 10^5^ conidia were inoculated in each one of the 50 mL-Erlenmeyer flasks filled with 25 mL of CDY. This medium is conducive for both *A. carbonarius* growth and OTA biosynthesis. Fungal growth was evaluated by weighing the mycelial part of *A. carbonarius* cultures, filtered through filter paper on a separatory funnel, after drying at 80 °C for 48 h.

### 4.6. OTA Extraction and Analysis, LOX Activity Determination

OTA was quantitatively extracted from cultural filtrate acidified by 1% of 0.1 M H_3_PO_4_ by using CHCl_3_ following the protocol described in [28]. The determination of LOX activity has been conducted following the procedure described in the same article.

### 4.7. Statistical Analysis

Data are presented as the value of mean ± SD of three independent measurements from three separate experiments. Mean values were evaluated using ANOVA and compared using Fisher’s Protected LSD Test (*p* ≤ 0.05).

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
