# Peer review of "Role of Some Food-Grade Synthesized Flavonoids on the Control of Ochratoxin A in Aspergillus carbonarius"

_molecules, 2019, doi:10.3390/molecules24142553_

Round 1

Reviewer 1 Report

The manuscript presents investigation of possible role of some food-grade flavonoids on the control of ochratoxin A in Aspergillus carbonarius, including their synthesis and NMR characterization . The subject of the manuscript is adequately reflected in the title; the most important information about the methods and results are summarized in the abstract. The authors neatly explained applied methods or provided references to some previous papers. Results are presented clearly, both in text and figures. Conclusions drawn from the results are without inconsistencies, justified and logical. References are of relevance for the subject of the investigation, providing up-to-date background and support for the obtained results.

Author Response

The authors thank the Reviewer for his words of appreciative remarks.

Reviewer 2 Report

The article is very important from the point of view of food safety because the development of toxinogenic fungi and contamination with ochratoxin A is still large. However, I have doubts as to whether it is appropriate to use synthesized compounds instead of natural compounds occurring in plants.

It would be advisable to carry out economic calculation of such a process in comparison with the use of biologically active compounds from plants. Moreover they often have completely different activities than the same compounds of chemical origin. In addition, it would be advisable to compare these two types of compounds. I do not see the possibility of their practical application for biocontrol. Can the authors specify this point. This can arouse consumer resistance, too.

In addition to the synthesis, two experiments were carried out in model media, assessing the amount of OTA and LOX activity after 3, 5 and 8 days with addition of the compounds in the amount of 5, 25 and 50 micrograms/ml. It would be advisable to carry out such an experiment in a natural medium, e.g. grape must.

I lack information on how synthetized compounds affect the rate of growth of Aspergillus carbonarius Given the growth of mold, I think that the assessment after 3 days is unnecessary. How much ochratoxin A forms the strain in the control sample, OTA amounts are given only as%.

I have doubts whether, after just 3 days, it reaches the maximum level of OTA synthesis (Figure 2).

In the methodology, the authors state that the amount of mycelium (line 263) has been studied, but there are no results.

Please explain what it means “AbsU” in Fig.2.

Author Response

The use of synthetic compounds has significantly reduced fungal contaminant growth and, in some cases, mycotoxin biosynthesis; however, their use also has resulted in resistant fungal isolates as well as problems relating to the compounds’ own toxicity and polluting potential.  Consequently, researchers have turned to extracts from plants that are edible or already known for beneficial health properties. Once compounds present in a biological active plant have been identified, their synthesis may be preferable to extraction and purification because it requires less solvent and time.  

In this study, synthesis of mosloflavone and negletein were found to be more advantageous than extraction.  However, in this case, a specific economic study was not conducted given the impossibility of accounting for all the relevant variables (e.g., different solvent costs, working hours, various location facilities). Thus, it would be interesting and useful to conduct an economic study that would define the conditions and circumstances most conducive to extractive and synthetic strategies, respectively. 

Concerning 5,6-dihydroxy-flavone, although this compound is not present in nature, its structure is highly correlated with those of negletein and mosloflavone; thus, we studied this molecule in order to evaluate the influence of the cathecolic structure in position C-5, C-6 in the absence of a methoxy group in position 7.  In sum, mosloflavone and negletein could be considered for practical applications, 5,6-dihydroxy-flavone is worth evaluating mainly to broaden knowledge of the effects of molecules with a similar structure.

The use of culture media containing appropriate material of natural origin (in this case, the grapes) enables use of an artificial, experimental system that is more similar to the natural system. However, at this stage of research, we preferred to conduct the experimental trials in a synthetic medium because this offered precise standardization of fungal growth curves as well as the start point and capacity of OTA production. These elements are less repeatable when the culture medium includes an inherently variable natural element. 

The tested compounds did not show any influence on fungal growth and consequently no graph was inserted in the paper, as noted in line 142-143: “None of the tested compounds had any effect on fungal growth (data not shown).” The amount of OTA synthesized by the fungus has been included in the Fig. 2 caption and in the text: “The amount of OTA found in the cultural filtrate of the untreated cultures was 7, 98, and 209 ng/mL after 3, 5, and 8 days respectively.” (lines 130, 131).

Although the OTA synthesized by the fungus after 3 days of incubation in the untreated samples is only 7 ng/mL cultural filtrates, it represents 100% of the fungal potential at that moment. 

In Fig. 3, the y-axis title “AbsU” means absorbance units/mg protein. The definition has been added in the caption of the figure and at line 146.

Reviewer 3 Report

This manuscript by Ricelli and coworkers establishes structure-activity relationships for flavonoid-mediated inhibition of ochratoxin A (OTA) biosynthesis and lipoxygenase (LOX) activity.  The synthesis of the 3 flavonoid compounds appears to have been carried out with care, and the derivatives are well-characterized by NMR (both 1H and 13C).  The results are interesting and should be of wide interest to the readership of Molecules.  This reviewer is in favor of publication pending minor editorial corrections.

1) In Schemes 1 and 2 it would be helpful to number the carbon atoms for 4, 5 and 7 so the reader knows exactly which carbon is 5 versus 6 etc.

2) The NMR for 7 was acquired in methanol-d4 in which it is not feasible to observe the phenolic hydrogens due to deuterium-exchange.  It would be better to acquire the spectra in DMSO-d6, the same solvent that was used for compound 5.  This way a direct comparison of the phenolic chemical shifts can be made.

3) Compound 4 (MOS) is a phenolic compound, while 5 and 7 are catechols.  The enhanced reactivity of 5 and 7 versus 4 will be due to their enhanced antioxidant potential and redox properties.  It should be possible to examine the literature and find oxidation potentials for these types of molecules.  This information would strengthen the Discussion section.  

Author Response

1) The carbon atoms have been numbered in chrysin-7-methyl-ether (1) (precursor to the synthesis) so as not to unduly clutter the scheme.

2) The spectrum of 5,6-dihydroxy-flavone 7 (acquired in methanol-d4) allows measurement of the coupling constant (J) of the two aromatic protons in both C-7 and C-8. This allowed us to affirm that the reaction correctly occurred in the C-6.  The literature reports 5,6-dihydroxy-flavone NMR spectrum in methanol-d4, so we also analyzed the NMR spectrum in the same solvent to compare our results with those previously reported.  

3) The authors agree that the comparison of redox potential among the compounds 4, 5, and 7 would be appropriate. Unfortunately, such information is not available in the literature and was impossible to obtain experimentally within a reasonable period. Further study including this aspect would be desirable.

Round 2

Reviewer 2 Report

I accept the explanations and corrections of the authors. The article in its current form can be published.